# Gestational Diabetes and Preterm Birth: What Do We Know? Our Experience and Mini-Review of the Literature

**DOI:** 10.3390/jcm12144572

**Published:** 2023-07-09

**Authors:** Agnesa Preda, Dominic-Gabriel Iliescu, Alexandru Comănescu, George-Lucian Zorilă, Ionela Mihaela Vladu, Mircea-Cătălin Forțofoiu, Tiberiu Stefaniță Țenea-Cojan, Silviu-Daniel Preda, Ileana-Diana Diaconu, Eugen Moța, Ioan-Ovidiu Gheorghe, Maria Moța

**Affiliations:** 1Department of Obstetrics and Gynecology, University Emergency County Hospital Craiova, 200642 Craiova, Romania; agnesapcela@yahoo.com (A.P.); dominic.iliescu@yahoo.com (D.-G.I.); alexcom8000@gmail.com (A.C.); zorilalucian@gmail.com (G.-L.Z.); 2Department of Obstetrics and Gynecology, University of Medicine and Pharmacy Craiova, 200349 Craiova, Romania; 3Department of Diabetes, Nutrition and Metabolic Diseases, Faculty of Medicine, University of Medicine and Pharmacy of Craiova, 200349 Craiova, Romania; ionela.vladu@umfcv.ro; 4Department of Diabetes, Nutrition and Metabolic Diseases, University Emergency County Hospital Craiova, 200642 Craiova, Romania; 5Department of Medical Semiology, Faculty of Medicine, University of Medicine and Pharmacy of Craiova, 200349 Craiova, Romania; 6Department of Internal Medicine 2, “Philanthropy” Clinical Municipal Hospital of Craiova, 200143 Craiova, Romania; 7Department of General Surgery, Faculty of Medicine, University of Medicine and Pharmacy of Craiova, 200349 Craiova, Romania; 8Department of General Surgery, C.F. Clinical Hospital, 200374 Craiova, Romania; 9Department of Surgery, University of Medicine and Pharmacy of Craiova, 200349 Craiova, Romania; sdpreda@gmail.com; 10Department of Pediatric Pneumology, “Marius Nasta” National Institute of Pneumophtisiology, 050159 Bucharest, Romania; dianadiaconu655@yahoo.com; 11Doctoral School, Faculty of Medicine, University of Medicine and Pharmacy of Craiova, 200349 Craiova, Romania; eugenmota@yahoo.com (E.M.); mmota53@yahoo.com (M.M.)

**Keywords:** gestational diabetes mellitus, preterm birth, maternal factors, pregnancy

## Abstract

Background: Gestational diabetes mellitus (GDM) is a form of diabetes that develops during pregnancy. The incidence of GDM has been on the rise in tandem with the increasing prevalence of obesity worldwide. We focused on the study of what causes premature births and if there are methods to prevent these events that can result in long-term complications. Methods: This study was a prospective, non-interventional study that lasted for 4 years from December 2018 to December 2022. From the group of women enrolled in the study, we selected and analyzed the characteristics of women who gave birth prematurely. Additionally, we performed a systematic review examining the association between GDM and the frequency of adverse pregnancy outcomes. Results: In total, 78% underwent an emergency caesarean and had polyhydramnios. The results indicate that women who had a preterm delivery had a significantly higher maternal age compared to those who had a term delivery (*p* < 0.001). Conversely, there was no significant difference in preconception BMI between the two groups (*p =* 0.12). Conclusions: In terms of the understanding of GDM and preterm birth, several gaps in our knowledge remain. The association between GDM and preterm birth is likely multifactorial, involving various maternal factors.

## 1. Introduction

Gestational diabetes mellitus (GDM) is a prevalent chronic condition during pregnancy, which poses a threat to the well-being of millions of women globally. Its recognition as a distinct entity was first established by O’Sullivan and Mahan [1,2].

GDM is a form of diabetes that develops during pregnancy and is characterized by elevated blood glucose levels and impaired insulin sensitivity. It is a transient condition that typically resolves after childbirth, but it poses significant health risks to both the mother and the developing fetus [3]. GDM occurs when hormonal changes during pregnancy lead to insulin resistance, resulting in inadequate insulin production and impaired glucose metabolism.

There are two main approaches to screening for GDM: universal testing, which evaluates all patients, and selective testing, which focuses on high-risk individuals. High-risk individuals include overweight or obese women with a history of giving birth to large babies, a family history of type 2 diabetes, or a personal history of gestational diabetes [4,5].

However, the ideal screening method remains unclear. The International Association of Diabetes and Pregnancy Study Groups (IADPSG) recommends a single-stage screening using a 75 g oral glucose tolerance test (OGTT) between 24 and 28 weeks of pregnancy [6]. Normal glucose values should be below specific thresholds: fasting—92 mg/dL (5.1 mmol), one hour—180 mg/dL (10 mmol), and two hours—153 mg/dL (8.5 mmol). If any of these values are higher, a diagnosis of gestational diabetes may be made.

This screening method is approved by both the World Health Organization (WHO) and the American Diabetes Association (ADA) [7].

ACOG (American College of Obstetrics and Gynecology) endorses a two-step approach, whereby screening for gestational diabetes mellitus (GDM) should initially be conducted using a 50 g glucose challenge test, and subsequently, if the screening test is positive, a 100 g oral glucose tolerance test (OGTT) should be performed [8].

GDM is managed through lifestyle modifications (medical nutrition therapy, physical activity) and, in some cases, drug therapy (insulin being recommended in most cases). Optimal glycemic control in GDM is crucial for minimizing the risk of maternal complications, promoting favorable fetal outcomes, and reducing the long-term health risks for both the mother and the child [9].

The incidence of GDM has been on the rise [3] in tandem with the increasing prevalence of obesity worldwide [10], resulting in an elevated risk of various pregnancy-related complications for affected women [11]. Accordingly, it is imperative to quantify the risk or odds of adverse pregnancy outcomes to facilitate appropriate preventative measures, risk assessments, and patient education.

It is widely acknowledged and supported by extensive documentation that women diagnosed with GDM are at an increased risk of developing type 2 diabetes mellitus (T2DM) in the years following pregnancy [12]. This association has been confirmed by various studies and is generally accepted as factual. However, the value of identifying and treating GDM has been a topic of heated discussion throughout the twenty-first century, with various issues arising [9,13].

GDM is linked to several negative perinatal outcomes and long-term consequences for both the mother and her child. These consequences include excessive fetal growth leading to macrosomia or infants that are larger than average for their gestational age [14], obstructed labor, and higher rates of operative deliveries, particularly via caesarean section [15]. Additionally, pregnancies complicated by GDM are more likely to result in preterm birth and hypertensive disorders of pregnancy, such as preeclampsia [16]. GDM is also associated with an increased risk of neonatal complications such as hypoglycemia, hyperbilirubinemia, hypocalcemia, polycythemia, respiratory distress syndrome, and birth trauma [17]. Moreover, children born to mothers with GDM are more prone to becoming overweight at an early age, developing obesity, T2DM, and cardiovascular diseases later in life [18]. Conversely, women diagnosed with GDM during pregnancy are at a higher risk of developing T2DM, metabolic syndrome, hypertension, and ischemic heart disease at a relatively young age, as compared to women who are not diagnosed with GDM [19].

Although extensive research has been conducted on maternal–fetal complications, our study primarily concentrates on the etiology of labor and premature birth, as well as potential preventive measures against these events, which necessitate intensive care for the newborn and may subsequently lead to enduring complications.

Women who have gestational diabetes during pregnancy are at an elevated risk of experiencing negative pregnancy outcomes. These can include preterm labor and an increased rate of fetal mortality [20]. It is crucial for these women to maintain adequate glycemic control during pregnancy to improve fetal and perinatal outcomes. The development of specialized maternal, fetal, and neonatal care has contributed to the decline in perinatal and neonatal morbidity and mortality rates for women with diabetes and their offspring. However, infants born to mothers with diabetes are still at risk of developing various complications, such as macrosomia, hypoglycemia, perinatal asphyxia, cardiac and respiratory problems, and birth injuries.

## 2. Materials and Methods

This study was a prospective, non-interventional study that lasted for 4 years from December 2018 to December 2022. We included 79 pregnant women in the study, who were monitored during pregnancy in the County Clinical Hospital Craiova and on whom we performed an oral glucose tolerance test (OGTT) with 75 g anhydrous glucose between weeks 24 and 28 of pregnancy. According to the obtained results, the pregnant women were diagnosed with GDM. The following plasma glucose levels were used to diagnose GDM: fasting plasma glucose: ≥92 mg/dL (5.1 mmol/L), 1 h plasma glucose: ≥180 mg/dL (10.0 mmol/L), 2 h plasma glucose: ≥153 mg/dL (8.5 mmol/L).

Inclusion criteria: aged over 18 years old, pregnant women who signed the study’s informed consent form and were monitored during pregnancy in the Emergency County Hospital, Craiova.

Exclusion criteria: women with type 1 and T2 DM prior to pregnancy, pregnant women aged less than 18 years, presence of other severe comorbidities (renal and cardiovascular diseases, anemia, thyroid disorders, cancer, etc.), or ongoing treatment that may have influenced the maternal and perinatal outcome, women who delivered outside the Emergency County Hospital, Craiova, patients who did not attend the 6-week follow-up visit to repeat the OGTT and reconfirm the diagnosis of GDM.

The study was conducted according to ethical principles and guidelines, with respect of the rights and well-being of study participants being essential in any research study. Adherence to the Helsinki Declaration and Good Clinical Practice guidelines helps ensure that ethical principles are followed and that the study is conducted with integrity and transparency. Providing informed consent and the right to withdraw from the study at any time gives participants autonomy and control over their involvement, and maintaining confidentiality protects their privacy.

From the group of women enrolled in the study, we selected and analyzed the characteristics of women who gave birth prematurely.

The distributions of continuous variables were tested for normality using the Kolmogorov–Smirnov test. Statistical analysis was conducted by using *t*-test, chi-square for percentages.

All tests performed were considered statistically significant if they recorded a *p*-value < 0.05.

## 3. Results

Premature birth occurred in 14 out of 79 (17.72%) pregnant women diagnosed with GDM during our study period. The general characteristics of women included in this group are shown in Table 1.

This table presents the general characteristics of pregnant women who gave birth prematurely and were diagnosed with GDM. The data are presented as averages with standard deviations (SD). The average age of the women was 32 years. They were hospitalized for an average of 26 days, had an average BMI of 26, and gave birth at an average gestational age of 33.36 weeks.

A high percentage of these women, specifically 78%, underwent emergency caesarean section delivery. Additionally, 78% of the women in this group had polyhydramnios, which is an excess accumulation of amniotic fluid in the uterus that can lead to complications during pregnancy and delivery. Finally, 50% of these women also suffered from hypertensive disorders during pregnancy. These findings suggest that women with GDM who give birth prematurely tend to have higher rates of caesarean delivery, polyhydramnios, and hypertensive disorders.

The fact that only 4 out of 14 women who gave birth prematurely and were diagnosed with GDM required insulin treatment during pregnancy suggests that lifestyle interventions, such as dietary modifications and increased physical activity, may have been effective in controlling blood glucose levels in most cases. Proper glycemic control during pregnancy is paramount for optimizing maternal and fetal outcomes. In our group, most of the fasting, one hour, and two hour postprandial glycemic values were within normal limits.

It is important to note, however, that some women may require insulin or other medications to achieve adequate glycemic control, and close monitoring of blood glucose levels is essential throughout pregnancy to ensure the best outcomes for both the mother and the baby.

After making a comparative analysis with the rest of the group, we obtained the following data, shown in Table 2.

This table presents the comparison between pregnant women with GDM who had a term delivery and those who had a preterm delivery. The table shows the mean values of maternal age, gestational age (GA) at delivery, and preconception BMI for both groups. The results indicate that women who had a preterm delivery had a significantly higher maternal age at delivery compared to those who had a term delivery (*p* < 0.001). Conversely, there was no significant difference in preconception BMI between the two groups (*p* = 0.12).

Out of 79 pregnant women with GDM, 14 delivered prematurely, with 11 (78.57%) undergoing an emergency C-section and only 3 (21.43%) delivering vaginally. On the other hand, out of 65 women who delivered at term, 60 (92.3%) had an emergency C-section, while only 5 (7.7%) had a vaginal birth. Overall, the rate of emergency C-sections was high in both groups.

We analyzed the occurrence of polyhydramnios and a normal amniotic fluid index (AFI) among pregnant women with GDM who gave birth either preterm or at term. Among those women who gave birth preterm, 11 (or 78.57%) had polyhydramnios, while 3 had a normal AFI. Of the 65 women who gave birth at term, 24 (or 36.92%) had polyhydramnios, while 41 had a normal AFI.

Analyzing the distribution of hypertensive and normal pregnancies among preterm and at-term deliveries shows that out of 14 preterm deliveries, 7 (50%) were hypertensive, while 7 (50%) were normal pregnancies. Compared with the group of pregnant women who gave birth at term, the data are not statistically significant (*p* = 0.209362).

Insulin treatment was higher among patients with a preterm delivery but did not reach statistical significance.

Overall, this table suggests that various factors, such as an increased incidence of polyhydramnios, could increase the risk of preterm delivery in pregnant women with GDM.

## 4. Discussion

The majority of women (78%) required an emergency C-section, which may suggest that these preterm deliveries were due to complications related to GDM. Glycemic imbalances in GDM patients can lead to acute fetal distress, jeopardizing the well-being of the unborn baby. In such cases, emergency fetal extraction via C-section becomes necessary to ensure the immediate rescue of the fetus from distress. Timely action is crucial to mitigate potential complications and promote the optimal outcome for both mother and child. However, adequate glycemic control is not sufficient to ensure fetal well-being, and the risk of fetal distress persists even in women with well-managed GDM. Therefore, the diabetogenic pathology associated with pregnancy can jeopardize fetal well-being independent of glycemic control and the type of treatment.

Higher C-section rates may be explained by the increased risk of fetal macrosomia, or excessive fetal growth, which can complicate vaginal delivery and increase the chances of birth injury. Additionally, 78% of these women had polyhydramnios. This condition is commonly seen in pregnancies complicated by GDM and can contribute to preterm labor and delivery [21]. Polyhydramnios can cause premature rupture of the membranes and preterm labor by putting pressure on the cervix, causing it to dilate prematurely. This pressure can also cause contractions of the uterus and increase the risk of preterm labor. Therefore, polyhydramnios is considered a risk factor for preterm birth and should be closely monitored during pregnancy. Polyhydramnios can also lead to abnormal fetal positioning, such as breech presentation, making vaginal delivery more challenging. Moreover, the increased risk of umbilical cord prolapse, placental abruption, and fetal distress further necessitate the need for C-section delivery in these cases.

Hypertensive disorders, such as gestational hypertension and preeclampsia, can have a significant impact on both maternal and fetal health during pregnancy. The table provided suggests that the prevalence of hypertensive disorders was higher in the preterm delivery group among pregnant women with GDM. This finding highlights the importance of the careful monitoring of blood pressure and early detection of hypertensive disorders in pregnant women with GDM, as it can potentially contribute to preterm delivery [22]. Further research is needed to explore the mechanisms behind this association and to develop effective preventive strategies for hypertensive disorders in pregnant women with GDM.

Regarding the relatively low use of insulin, this may be due to the temporary nature of GDM, which often resolves after childbirth. Moreover, dietary and lifestyle modifications are usually sufficient to manage blood glucose levels in most cases, with insulin therapy reserved for uncontrolled cases [23]. GDM is typically diagnosed later in pregnancy, resulting in a shorter duration of insulin treatment compared to other forms of diabetes.

With respect to the protracted duration of hospitalization for women with preterm labor (26 ± 19.91 days), it should be noted that in instances involving premature birth, the maternal hospital stay was likewise extended until the infant’s discharge within our institution. The concatenation of an extended stay in the neonatal intensive care unit and the prolonged hospitalization of the mother results in escalated healthcare expenditures. We performed a systematic review examining the association between GDM and the frequency of adverse pregnancy outcomes. The study adhered to the Preferred Reporting Items for Systematic Reviews and Meta-Analyses (PRISMA) guidelines and followed a predefined search strategy. Three databases, PubMed, Google Scholar, and Medscape, were searched, and inclusion and exclusion criteria were applied to identify relevant studies. Data extraction and quality assessment were conducted using standardized procedures.

The review was conducted by analyzing relevant studies published between January 2013 and May 2023. To ensure the inclusion of relevant studies, specific criteria were applied. Studies that compared pregnancy outcomes in maternal gestational were included. However, studies focusing on other forms of diabetes (e.g., maternal onset of diabetes in young, and type 1 and 2 diabetes), abstracts, review articles, letters, case reports, intervention studies, and non-English articles were excluded. Review articles were examined to identify any potential studies missed during the initial search.

Data extraction involved recording author details, study characteristics, sample sizes, population characteristics (including ethnicity), and pregnancy outcomes specific for women diagnosed with GDM. Two independent reviewers assessed the quality of the included studies. Disagreements between reviewers were resolved with the assistance of a third reviewer.

By following this rigorous methodology, the systematic review aimed to provide a comprehensive analysis of the association between GDM and adverse pregnancy outcomes. Definitions of various pregnancy outcomes were provided for clarity.

Caesarean section denotes the delivery of a fetus through a surgical incision made in the abdominal wall and uterus. Preterm birth is characterized by the delivery of a baby before the completion of 37 weeks of gestation [24]. Preeclampsia is defined as the onset of hypertension (blood pressure exceeding 140/90 mm Hg) and proteinuria (excretion of more than 0.3 g of protein in a 24 h urine collection) after 20 weeks of gestation [25]. Macrosomia refers to the condition where a baby is born with a weight exceeding 4000 g [26]. Stillbirth is the unfortunate event of fetal death occurring after 20 weeks of gestation or when the fetus weighs more than 500 g [27]. Large for gestational age (LGA) indicates a birth weight that exceeds the 90th percentile for gestational age [26].

This research identified 1650 articles, published between January 2013 and May 2023.

After eliminating duplicate articles and conducting a preliminary assessment of titles and abstracts, a total of 34 studies were subjected to further evaluation for eligibility. Subsequently, only 9 articles that met the inclusion criteria were selected for the review. The trial flow diagram is shown in Figure 1.

Nine articles published between 2013 and 2023 were included in the review. Six studies were retrospective, two were prospective, and one was a cross-sectional study.

Among the nine studies that analyzed data about gestational diabetes, the most frequently reported negative outcome was preterm birth (*n* = 9) followed by caesarean section (*n* = 7) (Table 3).

Caesarean section and preterm birth were the most frequently observed adverse outcomes. Other adverse outcomes included preeclampsia, macrosomia, stillbirth, AND LGA infants.

This table provides a concise overview of the adverse outcomes reported in various studies. It specifically focuses on caesarean section, preterm birth, preeclampsia, macrosomia, stillbirth, and LGA infants.

Multiple studies have reported caesarean section as an adverse outcome in women with GDM, with rates ranging from 27% to 50.23%. The higher rate from our institution may be explained by the lower data pool compared to the literature. Preterm birth occurred in 17.72% of women with GDM, falling within the literature-reported rates of 5.06–57.81%. Macrosomia was reported in 1.8% to 15.7% of cases, while LGA was reported in 3.4% to 14.4%. This outcome is also observed relatively frequently, suggesting it may have important clinical implications. Preeclampsia rates range from 2.6% to 12.9%, which may indicate that it is less commonly observed compared to C-section and preterm birth. Stillbirth variation was high, from 0.22% to an astonishing 46.10%, although the data derived from only two studies.

In summary, the table presents a clear overview of the adverse outcomes studied, indicating the relative frequency of their occurrence across different studies. However, it is important to note that further analysis and interpretation are necessary to fully understand the implications of these findings.

## 5. Conclusions

Despite significant advances in the understanding of GDM and preterm birth, several gaps in knowledge remain.

It is evident that preterm birth is a significant concern in women with GDM. The association between GDM and preterm birth is likely multifactorial, involving various maternal factors. It is a topic of great importance in the field of maternal–fetal medicine.

GDM is often associated with other comorbidities such as maternal obesity, hypertensive disorders, and maternal age, which are independently associated with an increased risk of preterm birth. These comorbidities may further compound the adverse effects on pregnancy outcomes, including preterm birth.

It is worth noting that the management of GDM, particularly through lifestyle modifications and medical interventions, can help mitigate the risk of preterm birth. Close monitoring of blood glucose levels, appropriate dietary management, regular physical activity, and, when necessary, insulin therapy are crucial in achieving optimal glycemic control and reducing the risk of adverse pregnancy outcomes.

Women with GDM are at an elevated risk of preterm birth compared to the general population. This increased risk may be attributed to the metabolic disturbances associated with GDM, the presence of comorbidities, and suboptimal glycemic control. Timely and comprehensive management of GDM, involving lifestyle modifications and medical interventions, is essential to minimize the risk of preterm birth and improve maternal and neonatal outcomes. Optimal multidisciplinary care involving obstetricians, endocrinologists, and nutritionists is necessary to achieve optimal outcomes.

While the exact mechanisms are still being unraveled, evidence supports a significant link between GDM and preterm birth. Recognizing the risk factors and implementing appropriate management strategies can help reduce the burden of preterm birth and its associated complications.

Continued research and clinical efforts are necessary to improve our understanding and management of this complex association and ultimately improve outcomes for women with GDM and their offspring.

## Figures and Tables

**Figure 1 jcm-12-04572-f001:**
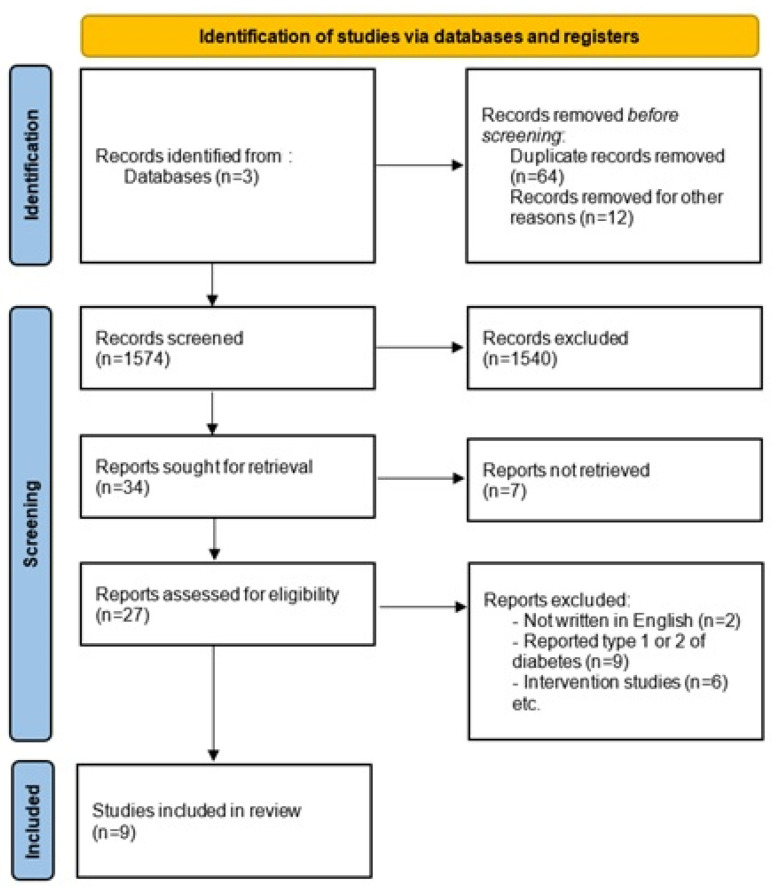
PRISMA 2020 flow diagram for reporting of systematic reviews. Delineation of study selection.

**Table 1 jcm-12-04572-t001:** General characteristics of pregnant women who delivered prematurely.

General Characteristics of Pregnant Women	Mean ± SD
Age (years)	32 ± 4.35
Days of hospitalization	26 ± 19.91
Preconception BMI (kg/m^2^)	26.48 ± 7.93
Gestational age at the moment of birth (weeks)	33.3 ± 2.58
Emergency C-section	78%
Polyhydramnios	78%
Hypertensive disorders	50%
Insulin treatment	28.57%

**Table 2 jcm-12-04572-t002:** Statistical comparison between the groups of women with GDM who gave birth at term and prematurely.

Pregnant Women with GDM	At Term Delivery	Preterm Delivery	*p*
Maternal age (years)	28.9 ± 2.90	32.36 ± 4.35	0.000313
Days of hospitalization	4.81 ± 1.52	26 ± 19.91	<0.00001
Preconception BMI (kg/m^2^)	23.63 ± 3.64	26.48 ± 7.93	0.12054
Emergency C-section	92.3%	78.57%	0.12226
Polyhydramnios	36.92%	78.57%	0.004433
Hypertensive disorders	32.30%	50%	0.209362
Insulin treatment	20%	28.57%	0.4816

**Table 3 jcm-12-04572-t003:** Pregnancy adverse outcomes reported in studies included in the review.

Study	Year	Number of Patients	C-Section % (*n*)	Preterm Birth % (*n*)	Macrosomia % (*n*)	Preeclampsia % (*n*)	Stillbirth % (*n*)	LGA % (*n*)
Darbandi et al. [20]	2021	140	46.81% (66)	57.81% (37)	8.7% (6)	ND *	46.10% (65)	ND *
Scott et al. [28]	2020	8609	49.7% (4278)	13.33% (1148)	ND *	12.9% (1114)	ND *	ND *
Reitzle et al. [29]	2023	283,210	37.6% (106,444)	7.4% (21,072)	ND *	ND *	0.22% (635)	14.4% (40,897)
Muhuza et al. [30]	2023	2048	27% (553)	9.62% (194)	6.74% (138)	ND *	ND *	3.4% (70)
Gou et al. [31]	2019	1523	50.23% (765)	5.06% (77)	10.57% (161)	7.09% (108)	ND *	12.41% (189)
Chung et al. [32]	2022	56	50% (28)	28.6% (16)	1.8% (1)	3.6% (2)	ND *	12.5% (7)
Billionet et al. [33]	2017	57,629	27.8% (16,021)	8.4% (4841)	15.7% (9048)	2.6% (1498)	ND *	ND *
Liu et al. [34]	2022	6786	ND *	21.46% (594)	ND *	ND *	ND *	ND *
Li et al. [35]	2023	16,806	ND *	5.18% (870)	ND *	ND *	ND *	ND *

* ND = no data; *n* = absolute number; % = percentage.

## Data Availability

The data presented in this study are available on request from the corresponding author.

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
