# Peer review of "Gestational Diabetes and Preterm Birth: What Do We Know? Our Experience and Mini-Review of the Literature"

_jcm, 2023, doi:10.3390/jcm12144572_

Round 1

Reviewer 1 Report

Dear Authors,

my comments:

1. all pregnancy complications must be defined properly, actually, they are not correct- polyhydramnios, macrosomia, preeclampsia.

2. A result that GA is lower in premature group is obvious, so that is not a result for me. 

3. In table 2- there is no comparison of insulin treatment in both groups. 

Author Response

Dear Reviewer,

We would like to express our gratitude for your invaluable contribution as a reviewer for our manuscript. Your insights have enriched the quality of our manuscript. We have addressed your comments accordingly.

Dear Authors,

my comments:

#1. all pregnancy complications must be defined properly, actually, they are not correct- polyhydramnios, macrosomia, preeclampsia.

Response: Thank you for the observation. We have reevaluated the definitions and citations and modified accordingly.

#2. A result that GA is lower in premature group is obvious, so that is not a result for me.

Response – this has been addressed and statement has been removed. Table data regarding lower GA in premature group has also been removed.

#3. In table 2- there is no comparison of insulin treatment in both groups. 

Response: We have added data for insulin treatment for both groups.

Reviewer 2 Report

The authors wrote a very interesting paper that combines a personal research with a review, and I want to congratulate them!

In the research substudy, I will suggest to comment more on the high incidence for C section.

In the review section, I will suggest that the authors should extend the presentation about the identified studies, giving information about the type of study, no of patients, duration, groups and obtained result.

Minor revision for spelling mistakes.

Author Response

Dear Reviewer,

We would like to express our gratitude for your invaluable contribution as a reviewer for our manuscript. Your insights have enriched the quality of our manuscript. We have addressed your comments accordingly.

The authors wrote a very interesting paper that combines a personal research with a review, and I want to congratulate them!

#1 In the research substudy, I will suggest to comment more on the high incidence for C section.

Response: The discussion section has been further updated according to your recommendations.

#2 In the review section, I will suggest that the authors should extend the presentation about the identified studies, giving information about the type of study, no of patients, duration, groups and obtained result.

Response: Table has been redesigned. We have included data regarding number of adverse events in patients with GDM, expressed as percentages and absolute numbers.

Reviewer 3 Report

The paper titles itself as a review, which in fact, is not. It's a mixture of several papers on GDM, given in a horrific mess, which does not give any glance on the topic by the reader.

The studied group has only 79 cases throughout four years!

The conclusions are really astonished: preterm delivery=lower GA!

78% of patients ended with c. section and polyhydramnios occurred. Authors stated that 50% of patients had hypertension, so the question is whether c.section has been performed because of GDM or because hypertension? Moreover, insulin treatment has been introduced in 28.5% so it's relatively low amount. Hospitalization was very long - average 26 days, which is not explainable.

Author Response

Dear Reviewer,

We would like to express our gratitude for your invaluable contribution as a reviewer for our manuscript. Your insights have enriched the quality of our manuscript. We have addressed your comments accordingly.

#1 The paper titles itself as a review, which in fact, is not. It's a mixture of several papers on GDM, given in a horrific mess, which does not give any glance on the topic by the reader.

Response: we would like to express our gratitude for your feedback. We hope that the revisions we have made have addressed your concerns and improved the overall quality of the article. We look forward to receiving any further feedback you may have and are committed to continually enhancing the clarity and value of our work.

#2 The studied group has only 79 cases throughout four years!

Response: We understand the importance of a robust sample size in research studies, and we would like to address this concern.

Firstly, we would like to clarify that our study represents a single center experience, which inherently limits the number of cases available for analysis. As a result, the sample size in our study is indeed smaller compared to multicenter studies. However, single center experiences can provide valuable insights and contribute to the existing body of knowledge, particularly in specialized areas. Although the number of cases in our study may be limited, it is essential to recognize that a single center study offers several advantages. For instance, it allows for greater homogeneity in terms of patient population, clinical practices, and data collection methods. This homogeneity enhances the internal validity of the study and reduces potential confounding factors that might arise from multicenter studies. It is important to note that due to the nature of our study design and the reliance on patients attending the follow-up visit, many patients were excluded from the study population. These exclusions primarily occurred when patients did not adhere to the recommended postpartum visit, thus preventing us from conducting the necessary tests to reconfirm the diagnosis of GDM. As a result, we were unable to definitively exclude preexistent diabetes in all cases. We would be happy to discuss any specific concerns or suggestions you may have regarding our study design or interpretation of results

#3 The conclusions are really astonished: preterm delivery=lower GA!

Response: This statement was retracted.

#4 78% of patients ended with c. section and polyhydramnios occurred. Authors stated that 50% of patients had hypertension, so the question is whether c.section has been performed because of GDM or because hypertension? Moreover, insulin treatment has been introduced in 28.5% so it's relatively low amount. Hospitalization was very long - average 26 days, which is not explainable.

Response: GDM and hypertension are not an indication to perform a Caesarian section. But both GDM and hypertension are confounders for C-section, as they both can lead to fetal distress. In the case of GDM, the condition is associated with an increased risk of fetal macrosomia, or excessive fetal growth, which can complicate vaginal delivery and increase the chances of birth injury. Polyhydramnios, characterized by an excessive accumulation of amniotic fluid, can lead to abnormal fetal positioning, such as breech presentation, making vaginal delivery more challenging. Moreover, the increased risk of umbilical cord prolapse, placental abruption, and fetal distress further necessitate the need for C-section delivery in these cases. Consequently, the coexistence of GDM and polyhydramnios poses additional obstetric complexities that often result in higher rates of C-sections for patients with these conditions. This has been further clarified in the manuscript.

Regarding insulin use, dietary and lifestyle modifications are usually sufficient to manage blood glucose levels in most cases, with insulin therapy reserved for uncontrolled cases. This has also been developed in the discussions section.

Regarding the long hospitalization, in cases with premature birth, the mothers hospital stay was also prolonged until the infant discharge. This has also been added to the manuscript.

Reviewer 4 Report

Comments

Line 57 - 58: “It is diagnosed through oral glucose tolerance tests, between 24 and 28 weeks of gestation, when fasting glucose levels are normal.” - This is not the only way to diagnose gestational diabetes. Clarify. Suggestion - you can use: Diabetes in pregnancy: management from preconception to the postnatal period NICE guideline [NG3 ]Published: 25 February 2015 Last updated: 16 December 2020 or some other guideline.

 Line 74 – 75:  “A point of debate was whether treating hyperglycemia in women with GDM effectively reduces the risk of adverse outcomes [9].”

Ref. 9 is: Yew TW, Chi C, Chan SY, Van Dam RM, Whitton C, Lim CS, et al. A Randomized Controlled Trial to Evaluate the Effects of a Smartphone Application–Based Lifestyle Coaching Program on Gestational Weight Gain, Glycemic Control, and Maternal and Neonatal Outcomes in Women With Gestational Diabetes Mellitus: The SMART-GDM Study. Diabetes Care. 2021 Feb 1;44(2):456–63.

The object of that study was to “examined whether Habits-GDM, a smartphone application (app) coaching program, can prevent excessive gestational weight gain (EGWG) and improve glycemic control and maternal and neonatal outcomes in gestational diabetes mellitus (GDM).”

The conclusion of that study was: “Habits-GDM (that integrated dietary, physical activity, weight, and glucose monitoring) resulted in better maternal glycemic control and composite neonatal outcomes (nonprespecified) but did not reduce excessive gestational weight gain EGWG among women with GDM.”

Cite properly or omit the sentence in line 74 – 75: “A point of debate was whether treating hyperglycemia in women with GDM effectively reduces the risk of adverse outcomes [9].”

It is well known that the treatment of GDM effectively reduces the risk of adverse outcomes (as you state at line 96-97 and line 173-174)..

Author Response

Dear Reviewer,

We would like to express our gratitude for your invaluable contribution as a reviewer for our manuscript. Your insights have enriched the quality of our manuscript. We have addressed your comments accordingly.

Comments

#1 Line 57 - 58: “It is diagnosed through oral glucose tolerance tests, between 24 and 28 weeks of gestation, when fasting glucose levels are normal.” - This is not the only way to diagnose gestational diabetes. Clarify. Suggestion - you can use: Diabetes in pregnancy: management from preconception to the postnatal period NICE guideline [NG3 ]Published: 25 February 2015 Last updated: 16 December 2020 or some other guideline.

Response: This issue has been further clarified in lines 58-75

#2 Line 74 – 75:  “A point of debate was whether treating hyperglycemia in women with GDM effectively reduces the risk of adverse outcomes [9].”

Ref. 9 is: Yew TW, Chi C, Chan SY, Van Dam RM, Whitton C, Lim CS, et al. A Randomized Controlled Trial to Evaluate the Effects of a Smartphone Application–Based Lifestyle Coaching Program on Gestational Weight Gain, Glycemic Control, and Maternal and Neonatal Outcomes in Women With Gestational Diabetes Mellitus: The SMART-GDM Study. Diabetes Care. 2021 Feb 1;44(2):456–63.

The object of that study was to “examined whether Habits-GDM, a smartphone application (app) coaching program, can prevent excessive gestational weight gain (EGWG) and improve glycemic control and maternal and neonatal outcomes in gestational diabetes mellitus (GDM).”

The conclusion of that study was: “Habits-GDM (that integrated dietary, physical activity, weight, and glucose monitoring) resulted in better maternal glycemic control and composite neonatal outcomes (nonprespecified) but did not reduce excessive gestational weight gain EGWG among women with GDM.”

Cite properly or omit the sentence in line 74 – 75: “A point of debate was whether treating hyperglycemia in women with GDM effectively reduces the risk of adverse outcomes [9].”

Response: This statement was removed.

#3 It is well known that the treatment of GDM effectively reduces the risk of adverse outcomes (as you state at line 96-97 and line 173-174).

Response: Thank you for this; we have removed the citation

Round 2

Reviewer 1 Report

Dear Authors,

I accept your response, but these definitions, which you deleted shoulb be presented in proper form in text.

Reviewer 3 Report

paper sufficient to be published but after inclusion all authors answers to the reviewers